# A Feature of the Crystalline and Amorphous Structure of Ultra Thin Fibers Based on Poly(3-hydroxybutyrate) (PHB) Containing Minor Concentrations of Hemin and a Complex of Tetraphenylporphyrin with Iron

**DOI:** 10.3390/polym14194055

**Published:** 2022-09-27

**Authors:** Svetlana G. Karpova, Ivetta A. Varyan, Anatoly A. Olkhov, Anatoly A. Popov

**Affiliations:** 1Department of Biological and Chemical Physics of Polymers, Emanuel Institute of Biochemical Physics, Russian Academy of Sciences, 4 Kosygina Street, 119334 Moscow, Russia; 2Academic Department of Innovational Materials and Technologies Chemistry, Plekhanov Russian University of Economics, 36 Stremyanny lane, 117997 Moscow, Russia

**Keywords:** hemin, TEMPO stable radical, correlation times, ultrathin fibers, poly(3-hydroxybutyrate), ozo-nation, amorphous phase

## Abstract

Comprehensive studies combining X-ray diffraction analysis, thermophysical, dynamic measurements by probe method and scanning electron microscopy have been carried out. The peculiarity of the crystalline and amorphous structure of ultra-thin fibers based on poly(3-hydroxybutyrate) (PHB) containing minor concentrations (0–5%) of a gene and a tetraphenylporphyrin (TFP) complex with iron (in the form of FeCl) are considered. When these complexes are added to the PHB fibers, the morphology of the fibers change: a sharp change in the crystallinity and molecular mobility in the amorphous regions of PHB is observed. When adding a gel to the fibers of PHB, a significant decrease in the degree of crystallinity, melting enthalpy, and correlation time can be observed. The reverse pattern is observed in a system with the addition of FeCl-TFP—there is a significant increase in the degree of crystallinity, melting enthalpy and correlation time. Exposure of PHB fibers with gemin in an aqueous medium at 70 °C leads to a decrease in the enthalpy of melting in modified fibers—to an increase in this parameter. The molecular mobility of chains in amorphous regions of PHB/gemin fibers increases at the same time, a nonlinear dependence of changes in molecular dynamics is observed in PHB/FeCl-TFP fibers. Ozonolysis has a complex effect on the amorphous structure of the studied systems. The obtained fibrous materials have bactericidal properties and should be used in the creation of new therapeutic systems of antibacterial and antitumor action.

## 1. Introduction

Since the diffusion of the drug substance is largely dependent on the structure of the polymer matrix, structural changes in the polymer can significantly affect the kinetics of drug release, and, consequently, the effectiveness of the dosage form. Structural changes in fibrillar mats during storage and operation can be caused by water absorption, heating, the action of oxygen, ozone and UV radiation on the polymer, as well as the action of microorganisms. These factors may act simultaneously or sequentially depending on the operating conditions of the dosage form and climatic features. One of the most common natural polyesters for medical use is poly(3-hydroxybutyrate) (PHB). An effective method of directed influence on the structure of a polymeric material is its doping with low molecular weight substances of various nature. In our earlier work, we considered the structure formation of fibrous materials based on PHB containing dipyridamole [1,2,3,4,5,6,7,8,9], chitosan [10], nanoparticles of titanium dioxide and silicon [11], complexes: iron (III)-chloroporphyrin [12], zinc porphyrin [13], manganese-chloroporphyrin [14], tin-chloroporphyrin [15]. A strong influence of low molecular-weight substances on the structure of the crystalline and amorphous phases of PHB fibers was shown. These substances, due to the presence of chemically active polar functional groups in them, enter into intermolecular interactions with the biopolymer. Depending on the nature of the metal, metalloporphyrins show a different tendency to aggregation, and this determines their ability to act as nucleating agents during polymer crystallization. In addition, porphyrins and their metal complexes have several binding centers in their structure, which contribute to the emergence of coordination interactions with the molecular environment [16,17]. First of all, the coordinating activity of the central-complexing metal ion should be singled out. Thus, metal cations with chloride extraligands contained in the structure of porphyrin metal complexes can exchange ligands for polar fragments of the environment, for example, polymeric, or such as oxygen-containing groups such as carboxyls or hydroxyls.

One of the most promising PHB dopants for the creation of polymeric materials for medical purposes are complexes of tetraphenylporphyrin (TPP) with tin (IV) (Sn(TPP)Cl_2_) and iron (III) (Fe(TPP)Cl). Sn(TPP)Cl_2_ is currently used to create photocatalysts that promote the destruction of organic toxicants and photosensitizers for medical diagnostics and therapy. A significant advantage of porphyrin complexes with tin(IV) is the presence of two extraligands directed on opposite sides of the plane of the porphyrin macrocycle. This structural feature causes the practical absence of the ability of these metal complexes to aggregate in comparison with the complexes of porphyrins with metals in the +2 and +3 oxidation states [18,19,20]. Fe(TPP)Cl molecules have a strong interaction with PHB macromolecules, however, the presence of only one extraligand chloride leads to intermolecular repulsion, so the molecules are capable of aggregation. Due to the unique geometric and electronic structure, the molecules of porphyrin metal complexes have a significant effect on the crystallization and segmental mobility of polymer macromolecules during the formation of composite matrices based on them. As a result of such an interaction, a sharp deceleration or acceleration of crystallization processes, a change in the structure of amorphous substances, and relaxation of polymer macromolecules can occur. Various complexes of porphyrins of natural and synthetic origin are used to modify the polymer to achieve specific properties of the entire composite system. Metal complexes with tetraphenylporphyrins have unique photocatalytic and antimicrobial properties [21,22].

Our aim was a comprehensive study of the properties of the new biocompatible composites based on a system of polymer and hemin for biomedical applications. One of the most promising areas for these materials is a wound-healing bandage: a biopoylmer-hemin-protein that provides regeneration. Also we found that there are different materials based on hemin: “hemin can be used in various biomedical materials, as a basis for binding proteins to a polymer, for container molecules (such as cavitands and capsules) for delivering systems, for constructing new biocatalysts tailored to specific functions, for creation of the innovative anticoagulants and others” [4,5,6].

In view of the fact that hemin is widely used in medicine as an independent drug complex in the treatment of porphyria, requiring a carrier polymer, and in various fields to create combined drugs based on proteins and peptides for drug delivery, combinations of PHB-hemin should definitely be recommended for use in biomedicine in view of the stability and high physical and mechanical properties of this composite.

The purpose of this work is to obtain new materials based on natural porphyrin—hemin and to compare the obtained regularities with the results obtained earlier for the PHB/Fe(TPP)Cl system. Due to its properties, hemin can be used in various biomedical materials, as a basis for binding proteins to a polymer [23], for container molecules (such as cavitands and capsules), for system delivery [24], to create new biocatalysts adapted for specific functions [25], which opens up prospects for the development of new biocompatible composites based on the system of nanopolymer and hemin for biomedical applications.

## 2. Materials and Methods

Polymer nanofibrous materials were obtained by electrospinning (ES) on a single-capillary laboratory setup with a capillary diameter of 0.1 mm. Polymer-molding solutions were prepared from 16F series semi-crystalline biodegradable polymer poly(3-hydroxybutyrate) (PHB) (BIOMER^®^, Germany) with a molecular weight of 206 kDa, a density of 1.248 g/cm^3^, and a crystallinity level of 59% (Figure 1a). Finely dispersed PHB powder was dissolved in chloroform at a temperature of 60 °C to obtain molding solutions. Hemin is an iron coordination complex (oxidation state: III) (Figure 1b), obtained by organic synthesis [26]. Hemin was dissolved in N,N-dimethylformamide at 25 °C and homogenized with PHB solution. The content of PHB in the solution was 7% wt.; the content of hemin was 1, 3, and 5% wt. The conditions of the ES process were voltage 17–20 kV, distance between electrodes 190–200 mm, and gas pressure on the solution 10–15 kg(f)/cm^2^.

X-ray diffraction analysis (XRD) of the samples was performed using transmission imaging. High-resolution two-dimensional scattering patterns were obtained using an S3-Micropix small- and wide-angle X-ray scattering system manufactured by Hecus (Cu(Ka) radiation, l = 1.542 Å). A Pilatus 100 K detector and also a PSD 50M linear detector were used in Ar/Me current at 8 bar argon pressure, high voltage on the Xenocs Genix source tube and current were 50 kV and 1 mA, respectively. The X-ray beam was formed using Fox 3D X-ray optics; the diameters of the forming slots in the collimator were 0.1 and 0.2 mm, respectively. The measurement range of diffraction angles was from 0.003 to 1.9 Å^−1^. To eliminate X-ray scattering in the air, the block of X-ray mirrors and the chamber were placed in a vacuum system (pressure ≈ 2.3 × 10^−2^ Torr). The accumulation time was varied in the range of 600–5000 s. This pattern of intermolecular interactions determined the structural and dynamic characteristics of ultrathin fibers. Hemin molecules have polar groups, which causes their attraction among themselves and, as a result, the formation of large particles.

Small-angle X-ray diffraction was used to study PHB/hemin fibers of various compositions. The average effective crystallite size *L_hkl_* in the crystallographic direction hkl was determined from the integral half-width of the line of the corresponding X-ray reflection using the Scherrer equation:(1)$$\Delta_{hkl}\left(2\theta \right)=\lambda /L_{hkl}\cos\theta$$
The value of the long period was calculated by the equation:(2)$$d=n\lambda /2\theta_{m}$$
where *d*—the long period, *λ*—1.542 Å wavelength of CuKα—radiation, $\theta_{m}$—diffraction angle, *n*—reflection order.

Electron paramagnetic resonance (EPR) X-band spectra were recorded on an EPR-V automated spectrometer (Federal Research Center for Chemical Physics, Russian Academy of Sciences, Moscow). The value of the microwave power to avoid saturation effects did not exceed 1 mW. The modulation amplitude was always much smaller than the resonance line width and did not exceed 0.5 G. The stable nitroxide radical TEMPO was used as a spin probe. The radical was introduced into the fibers from the gas phase at a temperature of 50 °C for an hour. The concentration of the radical in the polymer was determined by double integration of the EPR spectra; the reference was an evacuated solution of TEMPO in CCl_4_ with a radical concentration of ~1 × 10^−3^ mol/L.

The samples were studied by differential scanning calorimetry (DSC) on a DSC 204 F1 instrument (Netzsch) in argon at a heating rate of 10 K/min. The average statistical error in the measurement of thermal effects was ±3%. The enthalpy of fusion was calculated using the NETZSCH Proteus program. Thermal analysis was carried out according to the standard procedure [27]. Peak separation was carried out using the software “NETZSCH Peak Separation 2006.01”.

The fibers were obtained by electrospinning using a single-capillary laboratory setup with the following parameters: capillary diameter, 0.1 mm; electric current voltage, 12 kV; distance between electrodes, 18 cm; solution electrical conductivity, 10 μS/cm.

The effect on samples of distilled water was studied at 70 ± 1 °C. Before introducing the radical, the samples exposed to water were dried to constant weight.

The ozonation of the samples was carried out in the gas phase. The ozone concentration was 3 × 10^−5^ mol/L.

It should be noted that dopant molecules are localized only in amorphous regions of the polymer, and that the proportion of such regions decreases with the increasing concentration of the additive. For this reason, the local dopant concentration in the amorphous regions of the polymer was significantly higher than the set values of 1, 3 and 5%.

## 3. Results

### 3.1. Effect of Hemin Concentration on the Geometric Parameters of the Fiber

The PHB/hemin materials were obtained by double molding. The essence of the electrospinning process ensured the curing of the fibers of the polymer matrix after complete evaporation of the solvent [28]. This method made it possible to uniformly arrange additives that occupied the free space between crystallites in the amorphous regions of the polymer [29]. The introduction of hemin led to the formation of uniform and thin fibers, which might be due to the presence of metal, which increased the electrical conductivity of the spinning solution, and which, of course, improved the quality of the resulting fibers [30].

The introduction of hemin into the PHB structure accelerated the enzymatic hydrolysis of ultrathin PHB/hemin fibers compared to PHB/FeCl-TFP fibers due to a looser amorphous phase and a lower degree of crystallinity. When porphyrin metal complexes were added to the PHB molding solution, the morphology of the fibrous material changed dramatically. When 1–5% of FeCl-TFP complexes were added, elliptical elements in the fiber structure disappeared completely. When 1% of the FeCl-TFP complex was added, fibers were formed mainly with average diameters of 1.5–2.0; 3.0–4.0 and 5.0–6.0 microns. The presence of thin fibers of less than 3 microns was a consequence of the splitting effect of the primary jet of the molding solution in the field of electrostatic forces. With an increase in the concentration of FeCl-TFP from 3 to 5%, fibers with a diameter of 3 microns predominated.

Scanning electron microscopy was used to study in detail the morphology of fibrous materials. The images of the most characteristic parts of the samples are shown in Figure 1. The structure of the initial fibers of PHB and with a low content of hemin (up to 3%) is heterogeneous. The fibers are characterized by a high degree of tortuosity, adhesions and the presence of large formations in the form of thickenings of an elliptical shape. The average size of these structures is from 20 to 30 microns in the longitudinal direction and 15–25 microns in the transverse direction. An increase in the concentration of hemin leads to a decrease in the diameter and an increase in the uniformity of the fibers in diameter. The microrelief, with characteristic pores (Figure 2a) changes with the introduction of even 1% additive. Defects in the form of glues and thickenings are extremely rare, the fibers become thinner, the number of defects is reduced, and with the introduction of 5%, defects in the form of thickenings are almost completely absent. Roughness on the surface of the fibers almost completely disappears at 5% addition, as do elliptical thickenings.

The deepening of pores on such a thickening (Figure 2b) can be explained by more intensive evaporation of the solvent due to the technology of electrospinning using two solvents, which is used, among other things, to obtain highly porous fibers [30], however, for a 5% hemin content, this factor did not add to the contribution in the form of a more optimal balance of electrical conductivity-viscosity of the casting solution. In all likelihood, the presence of large hemin molecules in the intermolecular space of PHB led to an increase in the free volume of amorphous regions, which increased the rate of solvent desorption. To determine the uniformity of distribution of hemin in the fibers, we used the EDX method.

To ensure that the hemin was indeed incorporated into the fiber structure and not on the surface, an EDX analysis was performed. The results of the analysis of the composition of the material by the energy-dispersive X-ray method showed that hemin in the studied concentration range was distributed fairly evenly in the material, and not only concentrated in inclusions on the surface. Iron and chlorine atoms were chosen as the atom identifying hemin, since they were included in the central part of the tetrapyrrole ring.

### 3.2. X-ray Diffraction Study of PHB/Hemin Fibers of Different Composition

Along with a change in the geometry of ultrathin fibers with the introduction of an additive, one should also expect changes in the structural and dynamic parameters of the compositions.

It is known that the intermolecular interaction between porphyrin-metal complex particles, as well as between these particles and PHB molecules, depends significantly on the nature of the metal complex. Thus, it was shown in [31,32,33] that porphyrin molecules were attracted and, as a result, rather large particles of porphyrin were formed in PHB fibers. The intermolecular interaction of these particles with PHB macromolecules was extremely small. Although the molecules of the porphyrin complex with FeCl had a strong interaction with PHB macromolecules, the presence of the loride extraligand in the metal complex led to intermolecular repulsion, i.e., the molecules aggregated into smaller particles. Finally, Sn(TPP)Cl_2_ molecules were distributed in the system at the molecular level, due to the presence of two chlorine anions, repulsive forces between the complexes acted. The interaction of these complexes was weaker than that of Fe(TPP)Cl complexes with PHB macromolecules.

Figure 3a shows the dependence of the degree of crystallinity χ on the composition of the PHB/hemin system. For comparison, Figure 3b shows such a dependence for the PHB/Fe(TPP)Cl system. As seen, the dependencies point in different directions. Fe(TPP)Cl particles, being crystallization nuclei, caused an increase in the degree of crystallinity, and hemin, aggregating into large particles, prevented the crystallization of the PHB/hemin mixture and, as a result, the fraction of crystallites decreased with increasing hemin concentration in the polymer. In a system with 5% hemin, crystalline hemin particles 55–100 nm in size were formed. If the large period d increased with the addition of Fe(TPP)Cl particles to PHB (from 58 to 63 nm), then this parameter practically did not change in the PHB/hemin system.

It should be noted that with an increase in the concentration of Fe(TPP)Cl from 3% to 5%, a slight decrease in the degree of crystallinity was observed, apparently due to the aggregation of complexes into large particles, which did not lead to a further increase in the proportion of crystallites during fiber spinning.

Thus, X-ray diffraction study indicated a decrease in the degree of crystallinity in the PHB/hemin system, and in PHB/Fe(TPP)Cl fibers, an increase in the longitudinal dimensions (along the fibril) of crystal structures, a large period, and also the degree of crystallinity with increasing concentration of the additive was observed. Along with the presented changes, the transverse sizes of crystallites significantly decreased upon the addition of porphyrin: for example, L_020_ changed from 20.2 Å in PHB to ≈ 16 Å in PHB/Fe(TPP)Cl, and L_110_ changed from 20.1 Å in PHB to ≈14.5 Å in PHB with an additive. Therefore, it can be concluded that the addition of porphyrin to PHB fibers caused a decrease in the transverse dimensions of fibrils, and the longitudinal size of crystallites increased significantly. The observed changes, in our opinion, took place due to the straightening of the folded sections of the chains in the crystallites and their partial crystallization during the electrospinning of the fibers. The presence of small Fe(TPP)Cl particles contributed to the processes of intensive plasticization of the polymer, since it was in the PHB/Fe(TPP)Cl samples that the strongest increase in the degree of crystallinity occurred with increasing additive concentration. In PHB/hemin samples, however, the degree of crystallinity decreased significantly with an increase in the additive, which may be due to the formation of large particles, and with an increase in their concentration, the process of crystallite deformation took place to an increasing extent.

The presented data indicate that the addition of Fe(TPP)Cl particles to PHB fibers caused a significant increase in the degree of crystallinity with a higher longitudinal crystallite size (L_002_). When hemin was added, due to its aggregation into large particles, crystallization processes were inhibited, which led to a significant decrease in the degree of fiber crystallinity.

### 3.3. Thermophysical Characteristics of the Crystalline Phase of PHB/Hemin and PHB/Fe(TPP)Cl Compositions

Let us consider the effect of hemin on the crystalline phase of PHB fibers using the DSC method (Figure 3, Table 1). It should be noted that the concentration of the porphyrin complex in the amorphous regions of the polymer varied significantly beyond the specified values of 1, 3, and 5%. The latter were calculated for the entire mass of the sample, whereas the particles of the porphyrin complex were only in the amorphous interlayer, the proportion of which decreased significantly with an increase in the concentration of Fe(TPP)Cl and increased with the addition of hemin to the system. For example, in PHB fibers containing 5% Fe(TPP)Cl complex, its concentration in amorphous regions would be ∼15%.

Depending on the chemical structure of the additive, the indicators of ∆H and Tm varied greatly, due to the different interaction of PHB and the additive [34].

It is important to note that the enthalpy of melting ΔH obtained by DSC provided information both on the fraction of the crystalline phase and on linear structures (structures of straightened chains with a two-dimensional order). An increase in hemin concentration, aggregating into large particles, increasingly loosened the PHB structure, slowing down the process of crystallization and the formation of linear structures. As a result, ΔH decreased with increasing hemin concentration in PHB (Figure 3a). Moreover, there was an imbalance between the DSC and XRD data of ~40%, which was explained by the presence of linear systems in the polymer.

For PHB/Fe(TPP)Cl fibers, the opposite was observed. Having Fe(TPP)Cl particles of small size and being crystallization nuclei, due to a fairly strong interaction with PHB, they predetermined an increase in the degree of crystallinity and the proportion of linear structures in the composition (Figure 3b).

The thermograms of the PHB/hemin samples had an asymmetric shape with a low-temperature shoulder, which was due to the formation of linear structures. The general view of the thermograms of PHB/hemin samples subjected to repeated temperature scanning (heating) differed fundamentally from the thermograms of fibers of the same composition obtained during the first heating: single melting maxima transformed into bimodal maxima. We also observed a similar melting pattern for films and fibers with chitosan and dipyridomole [8,9,10]. The change in thermophysical characteristics during repeated temperature scanning was most likely associated with a deterioration in the organization of its crystalline phase due to a sufficiently high cooling rate when the fiber structure did not have time to return to its original state (the fibrous material passed into a film material).

### 3.4. Exposure of Ultrathin PHB Fibers in an Aqueous Medium in the Presence of Hemin and Porphyrin Complex Fe(TPP)Cl

Sorption and diffusion transport of water in PHB films were considered in detail earlier in a number of works, in which the diffusion coefficients and water absorption isotherms were estimated, the segmental mobility of PHB molecules was studied, and micrographs of the surface of samples obtained by SEM were presented, as well as their structural characteristics [15,35,36,37,38]. It was interesting to study the effect of exposure in an aqueous medium (distilled water) at a temperature of 70 °C on the structural and dynamic parameters of the PHB/hemin mixture composition and compare it with the regularities for the PHB/Fe(TPP)Cl system.

The structure and dynamics of medical materials to a large extent affect the diffusion processes of drugs introduced into polymers and, consequently, the kinetic patterns of drug release into the body’s environment. Since the polymer in the body is in the aquatic environment, it is important to identify changes in its structural and dynamic parameters during swelling. The choice of temperature for processing fibrous materials in water at 70 °C is associated with the need to increase the hydrophilicity of PHB, since after removing most of the water by drying, a smaller part of it remains in the polymer in a bound form (hydrogen bonds). Even a slight increase in the hydrophilicity of polymer matrices affects the diffusion processes of drug transport, which, as a rule, are polar. At room temperature, the diffusion of water into the polymer takes a long time, up to several days. The plasticizing action of water, which is most intense in the amorphous regions of the fibers, in combination with elevated temperature creates favourable conditions for the completion of additional crystallization of the polymer, leads to a more ordered state of the intergranular regions and to a redistribution of the ratio between the “loose” and dense phases. Earlier studies of the effect of an aqueous medium on the structure of polymers were carried out [12,14,36]. At 70 °C, hydrogen bonds are destroyed and the process of additional orientation of straightened chains in the fiber is realized. It was necessary to establish the degree of these structural changes for the system studied in the work.

Crystallites and amorphous regions generally do not correspond to the free energy minimum. The system tends to a minimum of free energy, and this process is facilitated when the intermolecular interaction decreases (for example: the structure loosens at elevated temperature, when additives are introduced into the polymer, etc.). Loose, amorphous regions relax to the state of a supercooled melt, and crystalline regions relax to a minimum of free energy through a change in size and, first of all, longitudinal dimensions (ΔG=ΔH−TΔS). It should be noted that amorphous regions in polymers are heterogeneous, there are regions with a high degree of straightening and regions with folded macromolecules. Under conditions where steric hindrance is removed (e.g., at elevated temperature), highly straightened chains assume more straightened conformations.

Water molecules act on the polymer structure in two ways: on the one hand, they have a plasticizing effect, which allows the crystallites to increase their size, on the other hand, after the removal of water, hydrated complexes remain in the fiber, loosening the polymer structure. Water molecules, hydrated complexes penetrate into the end surfaces of crystallites and linear structures, partially destroying them and, as a result, the degree of crystallinity decreases, as was shown in [15]. The accumulation of hydrated complexes in the amorphous component leads to its decompaction. It was necessary to establish the extent of these structural changes. At different exposure times in samples containing the addition of hemin and Fe(TPP)Cl at different concentrations, the intensity of the listed processes is different.

Let us first consider the effect of water treatment at 70 °C on the crystal structure of the PHB/hemin fiber (Table 1). Figure 4 presents the data on the change in the enthalpy of melting of this system on the concentration of hemin at different exposure times in an aqueous medium. An increase in ΔH in mixed compositions compared to the original untreated in an aqueous medium was observed at all studied times of water-temperature exposure to the polymer (with the exception of fibers with 5% hemin). In PHB fibers, the enthalpy of melting decreased. The research results indicated the contradictory nature of the impact of the aquatic environment on the fibers of PHB and PHB with complexes.

As previously noted, water molecules affected the polymer structure in two ways: on the one hand, they had a plasticizing effect, which allowed crystallites to increase in size. On the other hand, after the removal of water, hydrated complexes remained in the fiber, which are formed between the polar groups of PHB and hemin, as well as between hemin molecules, PHB groups. As a result, the fiber structure became looser and the enthalpy of the system decreased.

The increase in ΔH in PHB fibers with hemin after exposure can be explained by the predominance of plasticization processes and, as a consequence, an increase in the proportion of straightened chains in the fiber and, upon longer exposure, the formation of hydrated complexes that loosened the structure of crystalline regions (end surfaces of crystallites, linear structures). Hemin, due to large particle sizes, greatly loosens the structure of PHB, which leads to a significant concentration of water molecules in the fiber, the plasticizing factor also increases and, as a result, ΔH increases. In samples with 5% hemin, loosening was maximal and represented the highest increase in the melting enthalpy in them(Figure 5) It can be assumed that hemin, due to large particle sizes, does not deform the end surfaces of crystallites and linear systems; therefore, at a 5% additive in the polymer, the maximum opportunity for a certain proportion of macromolecules to straighten is realized; stress relaxation occurs in the nonequilibrium supramolecular structure of the fiber, which leads to additional densification of amorphous intercrystalline regions of their packing into crystallites.

Let us now consider the dynamic characteristics of the amorphous phase of ultrathin PHB/hemin fibers after water-temperature treatment. The plasticizing effect of water, which is most intense in the amorphous regions of the polymer, in combination with elevated temperature, created favorable conditions for the completion of its additional crystallization and for a more ordered state of the intercrystalline regions. After the removal of water, hydrated complexes remained in the fiber, which loosened the amorphous structure to a greater extent than non-hydrated particles; therefore, a sharp decrease in the correlation time was observed.

The specificity of this work was associated with a qualitative analysis of changes in the mobility of radicals during one or another treatment of the polymer matrix, and not with the interpretation of the exact values of the parameters of rotational mobility. For this reason, we needed to introduce a parameter that would qualitatively characterize the rotational mobility of radicals, be determined without a numerical analysis of all points of the spectrum, and have sufficient sensitivity to small changes in the shape of the spectrum, such as the changes shown in Figure 6. It can be seen that the greatest difference in the spectra was observed in the region of the high-field and low-field components; therefore, as the parameter of the spectrum shape, we chose the characteristic rotational correlation time calculated by the well-known formula [38,39]. This formula was proposed to calculate the rotational correlation times of nitroxyl radicals of the piperidine series in the case of their isotropic rotation in the region of rotational correlation times 5 × 10^−11^ ≤ τ ≤ 1 × 10^−9^ s. In this region, the EPR spectrum consisted of three well-resolved components, the width of which was described in terms of the Redfield theory. In our case, this parameter qualitatively characterized the line shape of the EPR spectrum and could be considered as some characteristic correlation time.

Figure 7a shows the change in the time of rotational correlation of radicals with an increase in the concentration of hemin complexes. The greatest intensity of the decrease in τ in the system was in the concentration up to 1% hemin. A further increase in the concentration of hemin in the fiber also caused an increase in molecular mobility, but not so significant. A similar pattern of change in χ and ΔH was observed with an increase in hemin concentration, which indicated the decompression of amorphous and crystalline regions upon the addition of hemin to the PHB structure.

In contrast to the observed changes in the structure of the amorphous component of the fiber with hemin, in the PHB/Fe(TPP)Cl system, in which the additive particles are small, opposite patterns of change in τ are observed (Figure 7b). Fe(TPP)Cl particles, being crystallization nuclei, promoted an increase in the degree of crystallinity and, as a result, a significant increase in the proportion of straightened macromolecules, which led to a decrease in molecular dynamics [12,35].

Let us now consider the effect of exposure to an aqueous medium at 70 °C on the molecular mobility of the PHB/hemin system. After 30 min exposure in an aqueous medium, τ increased for all systems, which indicated a slowdown in molecular mobility. However, with an increase in the time of water-temperature exposure, the mobility of chains in the amorphous phase increased significantly for all compositions of the system, although the enthalpy of melting remained higher than for untreated fibers. It can be assumed that as the density of amorphous regions (with increasing crystallinity) also increased, the concentration of the radical in them also decreased and, as a result, the molecular mobility increased. Characteristically, the most dramatic change in molecular dynamics and melting enthalpy was observed after the addition of 1% hemin at all times of water-temperature exposure. Apparently, the concentration of hemin increased not so sharply due to the formation of ever larger particles with an increase in the concentration of the additive.

The explanation of the obtained data was the fact that hemin, due to the large particle size, strongly loosened the PHB structure, which led to a significant concentration of water molecules in the fiber and to two opposite processes: not only the plasticizing factor increases, leading to the growth of straightened chains in the fiber, but to an increase in the concentration of hydrated complexes and, as a result, the structure of the amorphous phase was loosened. In our case, at all processing times, as shown by DSC and τ data (up to 30 min), the first factor prevailed. The decrease in the correlation time for all compositions of the system at treatment times of more than 30 min was explained by the ever lower concentration of the radical in the dense regions of the polymer, which were formed under water-temperature exposure, as well as by the loosening of the amorphous phase after the formation of hydrated complexes, the concentration of which increased with the time of fiber treatment. To investigate how the density of the fiber changed, data were obtained on the equilibrium concentration of the radical adsorbed in samples of the studied compositions of the same mass using the software from Bruker (winer). Figure 8 shows the dependences of the concentration of the radical (C) on the composition of the fibers for a series of exposure times. As expected, the concentration of the radical increased with increasing additive. After water-temperature treatment in PHB samples, the concentration increased with increasing exposure time, which was consistent with the increase in the fraction of the amorphous component according to DSC data. In fibers with hemin, the concentration of the radical decreased (deviation 3%). The addition of 5% hemin was accompanied by the highest increase in ΔH with increasing exposure time; the strongest changes in the radical concentration are observed precisely in these fibers. Therefore, it can be argued that despite the formation of hydrated complexes that loosened the polymer structure, the processes of plasticization of the structure leading to an increase in the enthalpy of melting, proceeding with the compaction of the material, prevailed.

Let us now consider the change in molecular dynamics under water-temperature treatment of PHB/Fe(TPP)Cl fibers. The dependence of molecular dynamics on the additive concentration after exposure to an aqueous medium was extreme. At an additive concentration of 1%, there was a sharp increase in τ. It was in these polymers that a high proportion of straightened chains was evidenced by a sharp increase in χ and Δ*H* with an increase in the concentration of the porphyrin complex. Therefore, at a low concentration of this additive, the plasticizing effect prevailes. As a result, mobility slowed down. At higher concentrations of Fe(TPP)Cl, the processes loosening the structure due to the formation of hydrated complexes began to predominate to an increasing extent, and the molecular mobility increased. Similar processes took place with an increase in the time of water-temperature treatment. The concentration of the radical in the fibers after exposure was significantly lower than in the original samples. For example, in the initial sample of PHB-porphyrin 5% concentration of the radical 9.3 × 10^15^, and after annealing—3.1 × 10^15^.

Thus, the hydrophilization of the studied systems depended both on the composition of the composition and on the type of additive in PHB, as well as on the time of water-temperature treatment. The sorption capacity, due to the high affinity of water for Fe(TPP)Cl and hemin molecules, increased sharply compared to PHB, in this case, the diffusion coefficients of water and low molecular weight compounds increased exponentially.

Studies of the structural and dynamic characteristics of fibrous materials PHB/hemin and PHB/FeClTPP after exposure to an aqueous medium at 70 °C showed that:The enthalpy of melting of PHB/hemin fibers at all times of water-temperature treatment (up to 5 h) and for all concentrations of hemin increased, which was due to the predominance of plasticization processes and structure compaction over loosening processes due to the formation of hydrated complexes in the polymer. The molecular mobility of the probe, and hence the mobility of macromolecules, slowed down after holding the fiber in an aqueous medium for 30 min due to an increase in the enthalpy of melting (according to DSC data). At higher times of such an impact on the system, the molecular dynamics began to increase, which was caused by the predominance of structure loosening processes during the formation of hydrated complexes. The radical was sorbed at a lower and lower concentration in amorphous regions with increasing exposure time in an aqueous medium due to an increase in density.Opposite regularities are observed when the water-temperature effect iwas applied to the PHB/Fe(TPP)Cl system. The presence of the complex determined the complex dependence of τ on its concentration in the fiber after exposure to water. The segregation of Fe(TPP)Cl molecules into small particles determined the structural and dynamic changes not only with an increase in the concentration of the additive, but also during water-temperature treatment of the polymer.

It is well known that water, penetrating into the polymer matrix, can affect both the physicochemical and mechanical properties of polymers and the flows of the third low molecular weight component (electrolyte, drug substance) [38].

Additional information on the dynamic behaviour of the PHB/hemin system of various compositions was obtained by studying the temperature dependence of the radical rotation rate and determining the corresponding activation energies E_τ_. A characteristic feature of E_τ_ as a function of the percentage of hemin in PHB was a sharp difference between these values in mixed compositions. This parameter decreased with increasing hemin concentration: 0%—50, 1%—28, 3%—22, 5%—18 kJ/mol. Such a sharp decrease in the activation energy of probe rotation upon the introduction of hemin was associated with a change in the state of the intercrystalline polymer phase; the density of the amorphous phase decreased significantly with an increase in the hemin concentration.

### 3.5. Effect of Ozonation on the Dynamics of Radical Rotation in PHB/Hemin and PHB/FeClTPP Composition Fibers

When biomedical materials are used, along with mechanical and thermal effects, their structure and segmental mobility are affected by ozone. Two sources of its appearance in the atmosphere should be indicated here. Firstly, ozone is formed during the operation of powerful electrical devices that ensure the vital activity of patients both during surgical operations and during therapy or monitoring in a hospital. Secondly, ozone in some special cases continues to be used in the sterilization of medical devices. Despite the daily contact of this aggressive compound with polymers, its effect on their morphological and dynamic characteristics remains a little-studied area of polymer materials science.

In the process of ozonation, two opposite processes take place that affect the molecular dynamics in the polymer: they are the breakage of macromolecules, which contributes to an increase in chain mobility and the formation of oxygen-containing groups on the side chains of macromolecules, which contributes to an increase in the rigidity of macromolecules and, as a result, a slowdown in molecular dynamics. The loops of macromolecules on the end surfaces of the crystallites are oxidized at the highest rate, since the highest stresses and, consequently, the highest oxidation rate are concentrated in these areas [12,35]. The rupture of these loops is accompanied by the straightening of regions of macromolecules, which slows down the molecular mobility in these regions. The formation of oxygen-containing groups in the side chains of macromolecules leads to an increase in intermolecular interaction and, as a result, the molecular mobility of the chains also slows down, whereas the rigidity of the chains increases and the processes of chain reorientation take place, which also leads to a slowdown in their mobility. In parallel, the processes of chain destruction proceed, which, as a result, can cause both the reorientation of chains and their folding into coils, depending on the initial degree of straightening of macromolecules. The tendency of a macromolecule to unfold and reorient itself or to coil into a ball is determined by the conformational criterion $k^{*}=h/L=\sqrt{2}/k$ (where h is the average distance between the ends of the chains, L is the contour length of the chain, k is the number of segments). Chains in which *k* > *k** tend to reorientate, macromolecules in which *k* < *k** tend to take the coil conformation.

Figure 8 shows the dependences of the rotational correlation time of the spin probe on the time of ozonation of the PHB/hemin and PHB/Fe(TPP)Cl systems. It can be seen that, firstly, the dependences were extreme, and, secondly, the change in the mobility of radicals with increasing ozonation time largely depended on the concentration of the additive and on the time of ozone-oxygen exposure of the polymer.

The extreme nature of the dependencies was due to the following factors. In PHB/hemin fibers (up to 30 min) and in PHB/Fe(TPP)Cl (up to 100 min), processes leading to a slowdown in molecular dynamics predominated. Moreover, if such changes in the PHB/hemin fiber were not so significant, then in the PHB/Fe(TPP)Cl system this effect was much greater, which indicated the predominance of processes leading to an increase in the rigidity of the polymer matrix. Let us consider the reason for the sharp difference in the patterns of change in molecular dynamics from the time of ozonation. The addition of Fe(TPP)Cl, as noted earlier, caused a significant increase in the degree of crystallinity and linear systems (these particles are the nuclei of crystallization), as a result, the proportion of straightened chains in the fiber greatly increased and, with an increase in the proportion of the composite, the χ and ΔH. The accumulation of oxygen-containing groups on such chains was accompanied by a significant increase in their rigidity and an increase in intermolecular interaction, as a result of which the coefficient k* increased and some of the macromolecules straightened, as a result, the molecular dynamics slowed down. Rupture of sufficiently straightened chains, which at the same time reorient themselves and their mobility also decreased. Another aspect leading to an increase in τ was the breaking and straightening of the chains on the end surfaces of the crystallites (due to the highest stresses on the chain folds). All these components caused a sharp increase in τ at the initial stage of ozonation of PHB/Fe(TPP)Cl fibers. With a longer exposure to ozone, destruction processes began to predominate with an increase in molecular mobility. With an increase in the concentration of the additive, the proportion of straightened chains, the proportion of crystallites increased, which caused an increasingly stronger growth effect τ with a change in the concentration of the additive (Figure 9).

In contrast to the Fe(TPP)Cl additive, hemin loosened the polymer structure to an increasing extent with increasing concentration. Against the background of a loose structure, the growth of intermolecular interaction during the accumulation of oxygen-containing groups did not have such a significant effect, and the fraction of end surfaces of crystallites also decreased with an increase in the fraction of hemin. As a result, a weak growth of τ is observed at the initial stage of ozonation. At deeper oxidation states, the molecular dynamics began to increase, and only after ozonation for more than 60 min did the chain mobility begin to decrease, which was apparently caused by the formation of a high concentration of oxygen-containing groups.

In general, the effect of ozone on the structural and dynamic characteristics of PHB doped with porphyrin metal complexes largely depended on the nature of the additive [15,36,38,40,41].

Thus, when comparing the results of ozonation of two polymers PHB/hemin and PHB/Fe(TPP)Cl, it follows that the structure that was formed upon addition of the complex determined to a large extent the ability of the system to change the dynamic parameters from the time of ozonation.

## 4. Conclusions

In this work, based on structural-dynamic studies, combining EPR, DSC and X-ray diffraction analysis, the results of the effect of low concentrations (1, 3, 5%) of hemin and the Fe(TPP)Cl complex on the degree of crystallinity, melting enthalpy and molecular dynamics of chains in amorphous regions of ultrafine fibers are presented. It is shown that the particle size of the additive and its chemical structure play an important role in the structuring of the fiber material. The intermolecular interaction between porphyrin complexes and the interaction of complexes with PHB macromolecules determine the degree of structural changes in the fiber. The addition of hemin to PHB fibers causes a strong loosening effect on both crystalline and amorphous regions of the polymer. The mutual influence of crystalline and amorphous regions in biodegradable highly crystalline polymers and their compositions remains a rather complex and little-studied problem of modern polymer materials science. These first studies have made it possible to interpret at the molecular level the effect of a number of aggressive factors (such as water-temperature and ozone effects) on the structural and dynamic characteristics of PHB–porphyrin complex fibers. Exposure to the aquatic environment and exposure to ozone in the gas phase greatly changes the structure of the fibers, and the observed changes depend on the nature and amount of the additive.

## Figures and Tables

**Figure 1 polymers-14-04055-f001:**
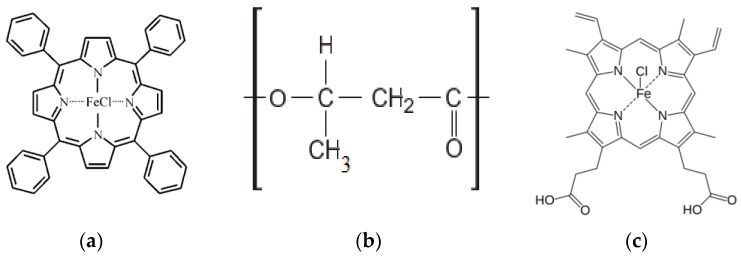
Structural formulas of (**a**) Fe(TPP)Cl, (**b**) PHB, and (**c**) hemin.

**Figure 2 polymers-14-04055-f002:**
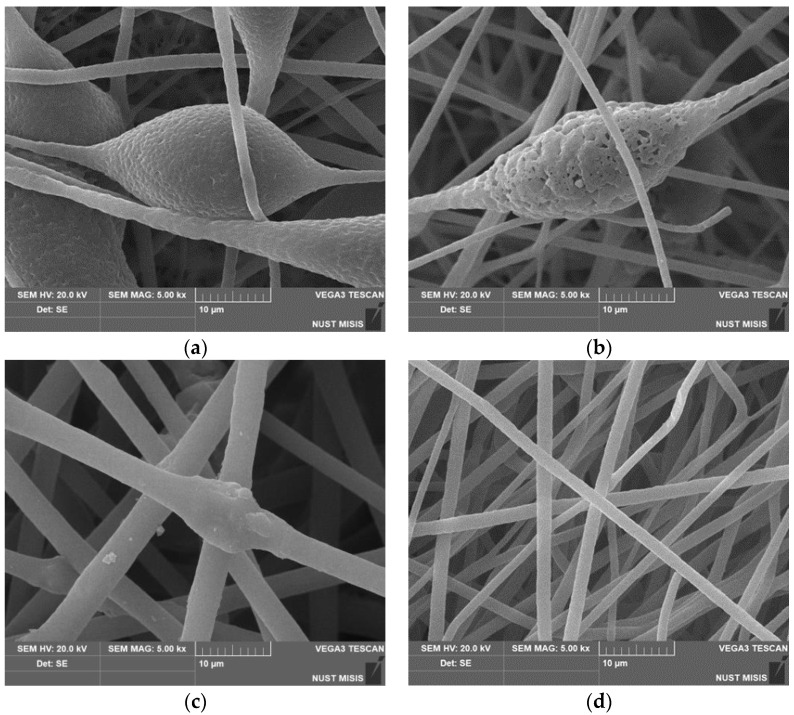
Micrographs of PHB with different hemin content: (**a**) 0%, (**b**) 1%, (**c**) 3% and (**d**) 5%.

**Figure 3 polymers-14-04055-f003:**
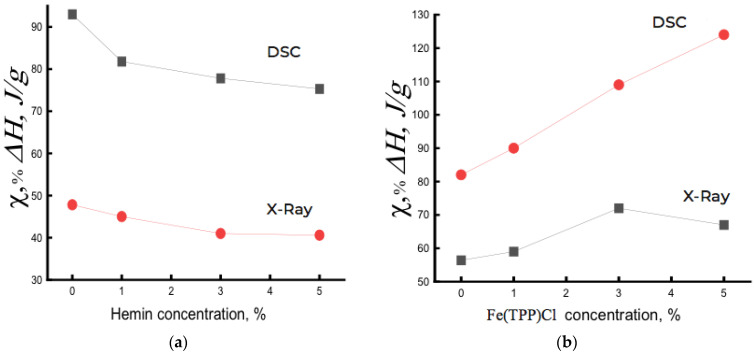
Dependence of the degree of crystallinity χ and enthalpy of melting ΔH on the composition of the mixed composition of (**a**) PHB/hemin and (**b**) PHB/Fe(TPP)Cl.

**Figure 4 polymers-14-04055-f004:**
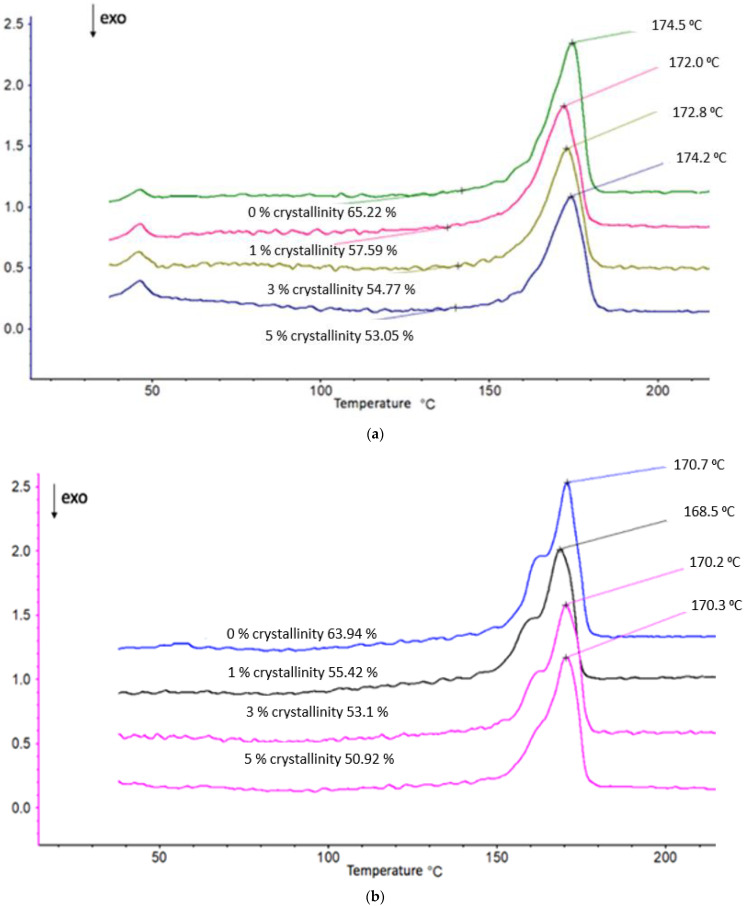
Thermograms of PHB/hemin melting (**a**) 1st and (**b**) 2nd scanning.

**Figure 5 polymers-14-04055-f005:**
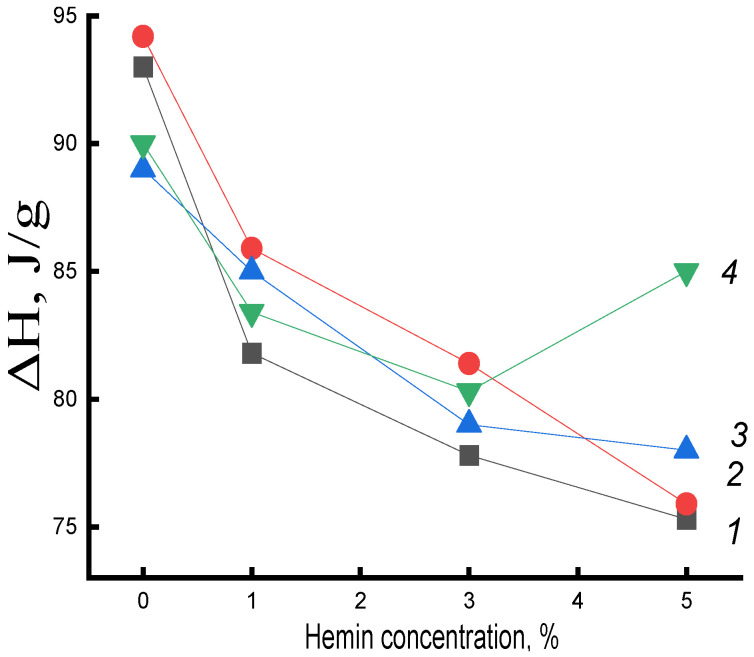
Dependence of the enthalpy of melting ΔH on the composition of the system for PHB fibers with hemin. Exposure time in the aquatic environment at 70 °C: 1—0, 2—30, 3—90, 4—300 min.

**Figure 6 polymers-14-04055-f006:**
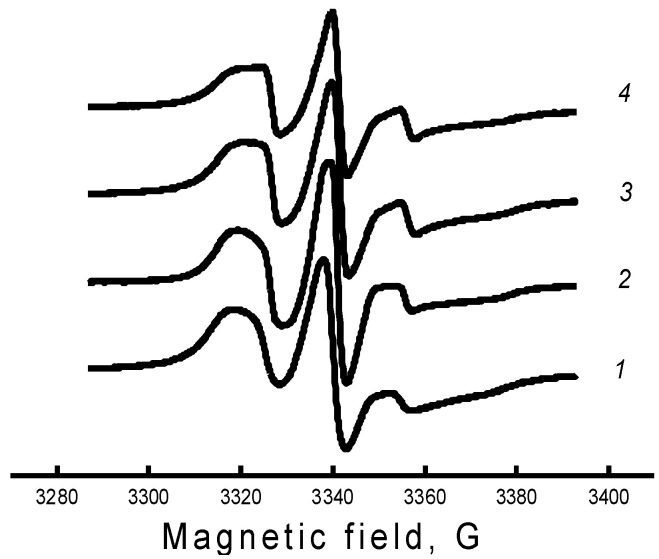
EPR spectra of the TEMPO nitroxide radical for PHB samples with hemin concentration: 1—0%, 2—1%, 3—3%, 4—5%.

**Figure 7 polymers-14-04055-f007:**
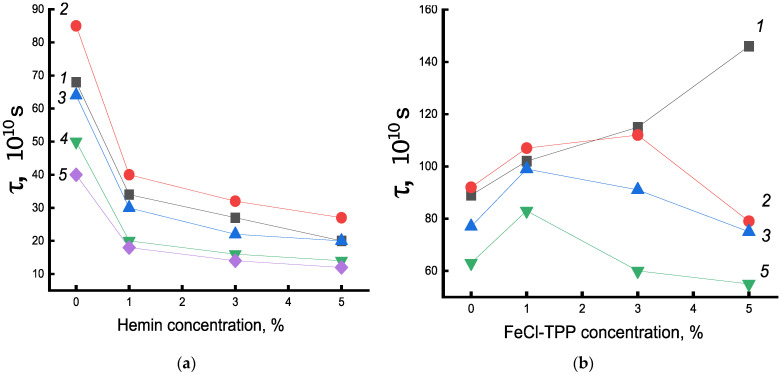
Dependence of τ on the composition of (**a**) PHB/hemin and (**b**) PHB/Fe(TPP)Cl. Exposure time in the aquatic environment at 70 °C: 1—0, 2—30, 3—50, 4—90, 5—300 min.

**Figure 8 polymers-14-04055-f008:**
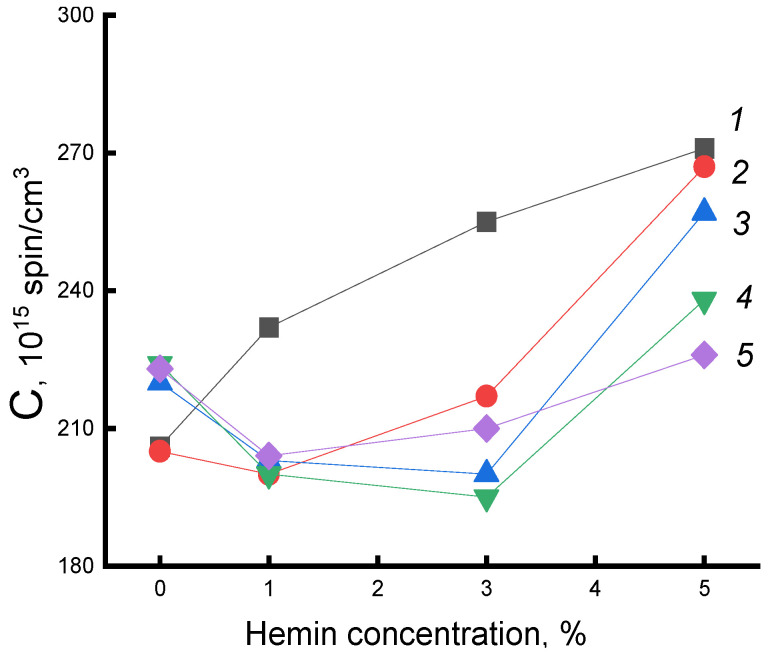
Dependence of the C radical concentration in PHB/hemin on the concentration of the additive. Exposure time in the aquatic environment at 70 °C: 1—0, 2—30, 3—50, 4—90, 5—300 min.

**Figure 9 polymers-14-04055-f009:**
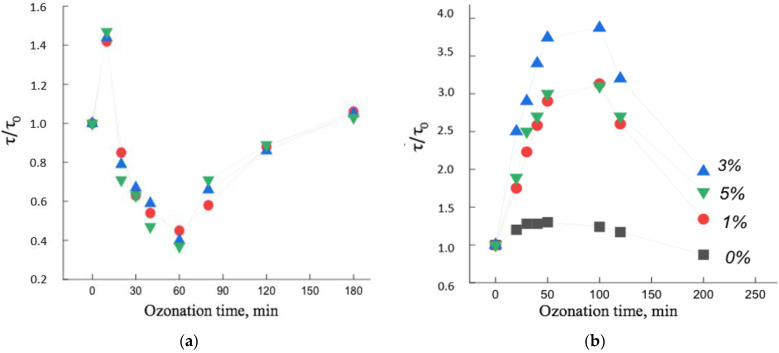
Change in molecular dynamics τ/τ_0_ as a function of ozonation time in PHB fibers with metal complexes: (**a**) hemin, (**b**) Fe(TPP)Cl: 1—0%, 2—1%, 3—3%, 4—5%.

**Table 1 polymers-14-04055-t001:** Enthalpy (ΔH) and melting point (T_m_) of ultrathin fibers of PHB/hemin mixtures studied by DSC.

Characteristic	PHB	PHB + 1%	PHB + 3%	PHB + 5%
ΔH, **1 scan**	93.05	81.8	77.8	75.3
ΔH, **2 scan**	90.8	78.7	75.3	72.7
T_m_ °C, **1 scan**	174.5	172	172.8	174
T_m_ °C, **2 scan**	170.7	168.5	170.2	170.3
**Annealing in an aqueous medium at 70 ** **°C for 0.5 h**				
ΔH, **1 scan**	88	84.5	81.6	67.3
ΔH, **2 scan**	81.3	74.8	76	64
T_m_ °C, **1 scan**	175	172	172.8	172.5
T_m_ °C, **2 scan**	171	160	172	172
**Annealing in an aqueous medium at 70 ** **°C for 1.5 h**				
ΔH, **1 scan**	89.2	84	76.7	74.3
ΔH, **2 scan**	77.8	87.7	78.3	77
T_m_ °C, **1 scan**	173.4	173.3	172.7	170.7
T_m_ °C, **2 scan**	168	170.3	172.7	168
**Annealing in an aqueous medium at 70 ** **°C for 5 h**				
ΔH, **1 scan**	90.2	80.7	80.3	81
ΔH, **2 scan**	92	75	80.9	78.9
T_m_ °C, **1 scan**	174.5	172.5	174	174
T_m_ °C, **2 scan**	168.5	166.4	171.7	170.5

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
