# Peer review of "A Feature of the Crystalline and Amorphous Structure of Ultra Thin Fibers Based on Poly(3-hydroxybutyrate) (PHB) Containing Minor Concentrations of Hemin and a Complex of Tetraphenylporphyrin with Iron"

_polymers, 2022, doi:10.3390/polym14194055_

Round 1

Reviewer 1 Report

In this paper, the authors examined ultra-thin fibers based on PHB containing minor concentrations (0-5%) of a gene and a TFP complex with iron. The experiments are not well organized and the results show very few important points affecting PHB properties’ improvement, and it is considered to be rejected. However, I have some questions and comments:

1.      Not well-prepared manuscript. The writing is so poor. The authors should improve this.

2.      There are a lot of mistaken sentences in this manuscript.

3.      Figure 4 is not so clear.

4.      In Fig 2, why 2a, 2B and 2C contained a bead on the fiber, but 2D look very smooth. Maybe problem occurred during the electrospinning operation with inconstant flowrate of polymer solution pump.

5.      Every graph has no statistical SD and/or at least 3 duplications. It should not be accepted at this moment.

Author Response

Thank you very much. We have corrected our manuscript. We turned to the proofreader for recommendations. We also tried to correct the most difficult to understand sentences on our own. We improved Figure 4. We also made corrections to the article.

The addition of hemin leads, in our opinion, to an increase in electrical conductivity and a decrease in the viscosity of molding solutions, which leads to an improvement in the geometry of the fiber (disappearance of thickening). Due to the unique geometric and electronic structure, the molecules of porphyrin metal complexes have a significant effect on the crystallization and segmental mobility of polymer macromolecules during the formation of composite matrices based on them. The introduction of hemin into the PHB structure accelerates the enzymatic hydrolysis of ultrathin PHB/hemin fibers compared to PHB/FeCl-TFP fibers due to a looser amorphous phase and a lower degree of crystallinity. When porphyrin metal complexes are added to the PGB molding solution, the morphology of the fibrous material changes dramatically. When 1-5% of FeCl-TFP complexes are added, elliptical elements in the fiber structure disappear completely. When 1% of the FeCl-TFP complex is added, fibers are formed mainly with average diameters of 1.5-2.0; 3.0-4.0 and 5.0-6.0 microns. The presence of thin fibers less than 3 microns is a consequence of the splitting effect of the primary jet of the molding solution in the field of electrostatic forces. With an increase in the concentration of FeCl-TFP from 3 to 5%, fibers with a diameter of 3 microns predominate. The disappearance of thickenings can be explained both by an increase in the electrical conductivity of molding solutions and by a decrease in the surface tension of the polymer solution when polar complexes are added. With an increase in the conductivity of the solution, the pulling of a drop of solution under the action of an electrostatic force occurs more intensively, which entails the alignment of the fibers in thickness.

The transition region of compositions of up to 1% porphyrin complexes, in which fibers with unstable geometry are formed, is common to all compositions of PHB molding solutions. Fibers with TFP molecules and their metal complexes are characterized by large values of the average diameter of cylindrical sections compared to PHB without additives. At the same time, polydispersity in diameters is observed. This indicates the effect of splitting the primary jet of the polymer solution during electroforming. The splitting effect can be associated with both an increase in the electrical conductivity of solutions and a decrease in the surface tension at the polymer-air interface.

An increase in the electrical conductivity of molding solutions can take place due to metal ions that are part of the TFP molecule. When TFP metal complexes are added to the molding solution, ionic conductivity occurs, leading to the leveling of the surface tension of the polymer solution droplet and the stability of the electroforming process. The relative decrease in the diameter of the fiber is correlated with the value of the specific electrical conductivity of the metal. Apparently, metal complexes with lower electrical conductivity interact to a lesser extent with the polar groups of PGB in the process of fiber formation, which leads to a more uniform maximum possible stretching of the jet and preferential orientation of the PHB macromolecules in the direction of extraction.

Thus, the introduction of TFP complexes leads to a change in the geometric parameters and spatial anisotropy of fibers in non-woven fibrous material. The change in the morphology of nonwovens is undoubtedly due to the heterogeneity of the supramolecular structure of the monofilament, which we further studied by X-ray diffraction analysis, DSC and EPR of the paramagnetic probe.

Metal complexes with tetraphenylporphyrins have unique photocatalytic and antimicrobial properties. It is shown that the particle size of the additive and its chemical structure play an important role in the structuring of the fiber material. The intermolecular interaction between porphyrin complexes and the interaction of complexes with PHB macromolecules determines the degree of structural changes in the fiber. The addition of hemin to the PHB fiber causes a strong loosening effect of both crystalline and amorphous regions of the polymer. While the addition of the FeCl-TFP complex causes the opposite effect – there is a significant increase in the degree of crystallinity, the enthalpy of melting and the correlation time of the probe. The conducted studies made it possible for the first time to interpret at the molecular level the effect of a number of aggressive factors (such as bottom-temperature and ozone effects) on the structural and dynamic characteristics of the fibers of the PHB-porphyrin complex. Exposure in the aquatic environment and exposure to ozone in the gas phase greatly changes the structure of the fibers, and the observed changes depend on the nature and amount of the additive.

Reviewer 2 Report

Recommendation: Minor revisions needed. 

Comments: 

The paper by Karpova et al. contributes to an analytical approach to assessing the properties of ultra-thin fibers based on PHB containing minor concentrations (0-5%) of a gene and a tetraphenylporphyrin (TFP) complex with iron. The article gives an interesting scientific perspective that when these complexes are added to the PHB fibers, the morphology of the fibers changes and a sharp change in the crystallinity and molecular mobility in the amorphous regions of PHB is observed. Some issues should be addressed prior to publication. 

  1. Page 4, lines 171-179. Why is there a different language used in this section? 

  1. Figure 2. Could you give a quantitative analysis of the average fiber diameter changes? What is the reason leading to these diameter changes?

  1. Page 6 line 224. Please provide the raw XRD images here or in the Supporting information.

  1.  
  1.  

Author Response

Thank you very much. We have fixed this error (lines 171-172).

The addition of hemin leads, in our opinion, to an increase in electrical conductivity and a decrease in the viscosity of molding solutions, which leads to an improvement in the geometry of the fiber (disappearance of thickening). Due to the unique geometric and electronic structure, the molecules of porphyrin metal complexes have a significant effect on the crystallization and segmental mobility of polymer macromolecules during the formation of composite matrices based on them. The introduction of hemin into the PHB structure accelerates the enzymatic hydrolysis of ultrathin PHB/hemin fibers compared to PHB/FeCl-TFP fibers due to a looser amorphous phase and a lower degree of crystallinity. When porphyrin metal complexes are added to the PGB molding solution, the morphology of the fibrous material changes dramatically. When 1-5% of FeCl-TFP complexes are added, elliptical elements in the fiber structure disappear completely. When 1% of the FeCl-TFP complex is added, fibers are formed mainly with average diameters of 1.5-2.0; 3.0-4.0 and 5.0-6.0 microns. The presence of thin fibers less than 3 microns is a consequence of the splitting effect of the primary jet of the molding solution in the field of electrostatic forces. With an increase in the concentration of FeCl-TFP from 3 to 5%, fibers with a diameter of 3 microns predominate. The disappearance of thickenings can be explained both by an increase in the electrical conductivity of molding solutions and by a decrease in the surface tension of the polymer solution when polar complexes are added. With an increase in the conductivity of the solution, the pulling of a drop of solution under the action of an electrostatic force occurs more intensively, which entails the alignment of the fibers in thickness.

The transition region of compositions of up to 1% porphyrin complexes, in which fibers with unstable geometry are formed, is common to all compositions of PHB molding solutions. Fibers with TFP molecules and their metal complexes are characterized by large values of the average diameter of cylindrical sections compared to PHB without additives. At the same time, polydispersity in diameters is observed. This indicates the effect of splitting the primary jet of the polymer solution during electroforming. The splitting effect can be associated with both an increase in the electrical conductivity of solutions and a decrease in the surface tension at the polymer-air interface.

Reviewer 3 Report

The manuscript by Karpova and co-authors presents very good interesting results on P(3HB) modification and its characterization. The authors could diversify P(3HB) composites characteristics, contributing to the PHA comunity and to broaden PHA use as biomaterials.

General comments:

1) For all the methods: please specify the model, supplier, etc of all the equipment and chemicals used in the study.

Specific comments:

L 15. Please clarify what you mean by 'a piece of a gene'.

L 51-53. A schematic figure indicating polymer structure and potential biding sites would contribute to readers understanding.

L 79. Please highlight the hemin properties that makes it suitable for the proposed applications.

L 88. The correct nomenclature is poly(3-hydroxybutyrate) - check IUPAC guidelines.

L 89. Did you experimentally obtained these values or it was informed by the manufacturer?

L 148-151. Please summarize data in a Table and compare with similar works (literature).

L 199-207. Should be moved to Materials and methods section.

L 172-179. Please correct.

Table 1. Usually manuscripts present the Tm from the second heating curve, since residual solvents or other contaminants are released during the first heating cycle. Do you have a reason to present Tm obtained on both 1st and 2nd heating cycles?

L 262-266. Please compare your ∆H and Tm results with other PHA-based materials, such as P(3HB-co-3HV) (https://doi.org/10.1021/acs.biomac.0c00826), P(3HB-co-3HHx) (https://doi.org/10.1016/j.ijbiomac.2022.06.024) for example.

L 273-277. Please supply the original thermograms from your equipment as supplementary material.

L 297-298. Clarify 'aquatic environment'

Figure 7, Figure 8, and Figure 9. Please present standard deviation and statistical significance for the presented data.

L 573-575. Based on your results and discussion, can you suggest possible applications for the obtained materials?

L 592-594. Should be moved to Materials and Methods section.

Author Response

Thank you very much. We wrote in the introduction: “Our aim was a comprehensive study of the properties of the new biocompatible composites based on a system of polymer and hemin for the biomedical application. One of the most promising areas for these materials is a wound-healing bandage: biopoylmer-hemin-protein that provides regeneration.” Also we found that there are different materials based on hemin: “hemin can be used in various biomedical materials, as a basis for binding proteins to a polymer, for container molecules (such as cavitands and capsules) for delivering systems, for constructing new biocatalysts tailored to specific functions, for creation of the innovative anticoagulants and others” In view of the fact that the hemin is widely used in medicine, as an independent drug complex in the treatment of porphyria, requiring a carrier polymer, and in various fields of creating combined drugs based on proteins and peptides for drug delivery, combi-nations of PHB-hemin should definitely be recommended for use in biomedicine in view of the stability and high physical and mechanical properties of this composite.” So the future purpose of our work is the creation of a bandage for regenerative medicine. It is possible because there are various works and a number of studies confirm the effectiveness of attachment of proteins for wound healing to hemin and other metalloporphyrins containing atom of Fe. But only the hemin of whole of the entire line of metalloporphyrins has all the listed advantages in the case of creating a composite material based on a biocompatible polymer and has a natural origin.

Point 4: L 88. The correct nomenclature is poly(3-hydroxybutyrate) - check IUPAC guidelines.

Response 4: Thank you very much. We have fixed this error.

Point 5: L 89. Did you experimentally obtained these values or it was informed by the manufacturer?

Response 5: This data was provided to us by the manufacturer.

Point 6: L 148-151. Please summarize data in a Table and compare with similar works (literature).

Response 6: Thank you for your suggestion. We have added this information

Point 7: L 199-207. Should be moved to Materials and methods section.

Response 7: Thank you very much. We have fixed this error

Point 8: L 172-179. Please correct.

Response 8: Thank you very much. We have fixed this error

Point 9: Table 1. Usually manuscripts present the Tm from the second heating curve, since residual solvents or other contaminants are released during the first heating cycle. Do you have a reason to present Tm obtained on both 1st and 2nd heating cycles?

Response 9: We are grateful for Your comment. Since in our case the fiber material, after the first scan, the structure has changed dramatically and therefore the first scan gives true information about the enthalpy of melting.

Point 10: L 262-266. Please compare your âˆ†H and Tm results with other PHA-based materials, such as P(3HB-co-3HV) (https://doi.org/10.1021/acs.biomac.0c00826), P(3HB-co-3HHx) (https://doi.org/10.1016/j.ijbiomac.2022.06.024) for example.

Response 10:

Depending on the chemical structure of the additive, the indicators of ∆H and Tm vary greatly, due to the different interaction of PHB and the additive. We have added this information to our article.

Point 11: L 297-298. Clarify 'aquatic environment'

Response 11:

Thank you very much. We have fixed this error. The study was conducted in distilled water.

Point 12: Figure 7, Figure 8, and Figure 9. Please present standard deviation and statistical significance for the presented data.

Response 12: We are grateful for Your remark. We have added this information.

Point 13: L 573-575. Based on your results and discussion, can you suggest possible applications for the obtained materials?

Response 13: So the future purpose of our work is the creation of a bandage for regenerative medicine. It is possible because there are various works and a number of studies confirm the effectiveness of attachment of proteins for wound healing to hemin and other metalloporphyrins containing atom of Fe. But only the hemin of whole of the entire line of metalloporphyrins has all the listed advantages in the case of creating a composite material based on a biocompatible polymer and has a natural origin.

Point 14: L 592-594. Should be moved to Materials and Methods section.

Response 14: Thank you very much. We have made corrections
